# Distinct African Swine Fever Virus Shedding in Wild Boar Infected with Virulent and Attenuated Isolates

**DOI:** 10.3390/vaccines8040767

**Published:** 2020-12-16

**Authors:** Aleksandra Kosowska, Estefanía Cadenas-Fernández, Sandra Barroso, Jose M. Sánchez-Vizcaíno, Jose A. Barasona

**Affiliations:** 1VISAVET Health Surveillance Center, Complutense University of Madrid, 28040 Madrid, Spain; estefania.cadenas@ucm.es (E.C.-F.); sandrabarroso@ucm.es (S.B.); jmvizcaino@ucm.es (J.M.S.-V.); 2Department of Animal Health, Faculty of Veterinary, Complutense University of Madrid, 28040 Madrid, Spain

**Keywords:** African swine fever, wild boar, shedding patterns

## Abstract

Since the reappearance of African swine fever virus (ASFV), the disease has spread in an unprecedented animal pandemic in Eurasia. ASF currently constitutes the greatest global problem for the swine industry. The wild boar (*Sus scrofa*) in which the pathogen has established wild self-sustaining cycles, is a key reservoir for ASFV, signifying that there is an urgent need to develop an effective vaccine against this virus. Current scientific debate addresses whether live attenuated vaccines (LAVs), which have shown promising results in cross-protection of susceptible hosts, may be feasible for vaccinations carried out owing to safety concerns. The objective of this study was, therefore, to compare the ASFV shedding in wild boar infected with virulent and attenuated (LAV) isolates. Different shedding routes (oral fluid and feces) and viremia rates were characterized in wild boar inoculated with Lv17/WB/Rie1 isolate (*n* = 12) when compared to those inoculated with the virulent Armenia07 isolate (*n* = 17). In general, fewer animals infected with the Lv17/WB/Rie1 isolate tested positive for ASFV in blood, oral fluid, and feces in comparison to animals infected with the virulent Armenia07 isolate. The shedding patterns were characterized in order to understand the transmission dynamics. This knowledge will help evaluate the shedding of new LAV candidates in wild boar populations, including the comparison with gene deletion mutant LAVs, whose current results are promising.

## 1. Introduction

African swine fever (ASF) is a viral hemorrhagic disease that affects domestic and wild suids. ASF causes a wide spectrum of clinical pictures characterized by high fever, anorexia, lethargy, and generalized hemorrhages, and it is, in most cases, almost 100% lethal in domestic pigs and Eurasian wild boar (*Sus scrofa*) [1]. ASF virus (ASFV) is endemic to sub-Saharan African countries, along with some countries in the Transcaucasia region [2]. ASFV genotype II has been circulating in Eastern Europe since 2007, spread to the European Union in 2014, and from there to Asia in 2018 [2]. ASFV has shown a capacity for rapid and far-flung spread in Asian countries, triggering numerous outbreaks with a serious socioeconomic impact on the countries affected [3]. Once the infection reaches a country, the consequences are devastating, not only because of the significant impact on animal health, but also owing to the enormous economic losses derived from the slaughter of affected and in-contact animals and trade restrictions. There is, therefore, a global urgency to develop a safe and effective ASF vaccine.

Studies with live attenuated vaccines (LAVs) obtained from naturally occurring low-virulence ASF strains (OURT88/3 or NH/P68) have shown promising results with regard to protecting domestic pigs against the homologous virus, and they have provided partial cross-protection against heterologous strains. Another example of an LAV candidate is the Lv17/WB/Riel ASFV isolate, which was proven to provide over 92% protection against challenge with a virulent ASFV isolate (Arm07) in domestic and wild suids [4,5]. However, practical experience has revealed numerous safety and efficacy concerns about these kinds of vaccine candidates, which are derived from their inherent infectious nature, such as the shedding of the attenuated live virus, reversion to virulence, or recombination between field and attenuated strains, among others [6].

ASFV is generally disseminated by means of direct contact between infected and healthy animals or contact with infected excretions and secretions. Indirect transmission can also occur through the ingestion of contaminated meat or contact with various contaminated fomites, in which ASFV remains infectious for long periods [7]. In this respect, shedding routes and intensity define the probability of pathogen transmission through direct and indirect contact [8]. It is, therefore, necessary to evaluate the shedding patterns of different ASFV isolates, especially in the main wild reservoir hosts, in which proper surveillance and disease control schemes are more complex, as is the case of wild boar (*Sus scrofa*) in the current ASF epidemy [9]. Wild boar populations have undergone an unprecedented demographic explosion in their native historic range in the last 35 years, from Western Europe to Eastern Asia [10,11]. This demographic trend has occurred along with a significant increase in direct or indirect contact, pathogen transmission, and geographical distribution [12]. These epidemiological changes may also increase the risk of transmission of ASFV at the wild boar–domestic pig interface [13].

Previous infection studies on wild boar with high-virulence strains have shown that virus shedding starts after a latent period of between 2 and 6 days [14]. In the case of moderate- and low-virulence isolates, this period ranges from 1 to 7 days [15,16,17], and, in some cases, excretions from animals infected with low-virulence strains have remained infectious for more than 70 days [15]. These excretions could consequently be disseminated in the environment and act as a source of infection [18]. Of all the possible shedding routes of ASFV, oropharyngeal would appear to be the most common path [15], although rectal fluids also involve a considerable risk, since infected pigs could have hemorrhagic enteritis, thus leading them to eliminate highly infected blood through feces [19]. Another important route of transmission in wild boar is their blood, as these animals are often involved in fights and practice cannibalism [20]. Furthermore, several of the routes studied, such as nasal, ocular, and vaginal routes, have proven to have consistently lower titers [15,21]. It was for this reason that blood, oral fluids, and feces were selected as shedding routes for this study.

This paper examines the shedding patterns of the attenuated ASFV isolate Lv17/WB/Riel in order to evaluate its potential as a vaccine candidate for wild boar. The possible survival and shedding of AFSV were investigated by collecting blood, oral swabs, and feces while carrying out previously published studies in which wild boars were orally inoculated with the Lv17/WB/Rie1 ASFV isolate [5] and another study where wild boars were intramuscularly inoculated with the highly virulent Arm07 ASFV isolate [22]. This information will help to determine the potential risk of the deliberate release into the environment of this attenuated isolate in comparison with highly virulent field isolates and will additionally complement studies previously described by Barasona et al. [5] and Rodríguez-Bertos et al. [22].

## 2. Materials and Methods 

### 2.1. ASFV Isolates

The attenuated, non-hemadsorbing, p72 genotype II ASFV Lv17/WB/Riel isolate was used for immunization purposes. This isolate was previously described and tested in both domestic pig and wild boar in protection trials [4,5]. The results from these trials have demonstrated the great potential of this isolate, showing a high level of protection against the challenge with a highly virulent virus in wild boar (92%) [5]. The virus was grown for 7 days in porcine blood monocytes (PBM) [23] and was collected as described previously in [5]. Viral titer was defined as the amount of virus causing cytopathic effects in 50% of infected cultures (TCID_50_/mL), as estimated through the use of immunoperoxidase staining.

The hemadsorbing p72 genotype II ASFV Armenia07 isolate was used as the field virulent virus. This isolate was previously described as a highly virulent ASFV in wild boar, causing an acute rapid progression of the disease [22]. The virus was propagated in PBM as described in [24]. Viral titer was defined as the amount of virus causing hemadsorption in 50% of infected cultures (HAD_50_/mL/TCID_50_/mL).

### 2.2. Animals

We evaluated a total of 29 animals originating from two previous independent experiments. All these animals were female wild boar piglets that were 3–4 months old and weighed 10–15 kg, which were obtained from a commercial wild boar farm in Extremadura, Spain. The experiments were performed under biosafety level 3 conditions at the VISAVET Centre at the University Complutense of Madrid. The animals were acclimated 2 weeks before the experiments began. During the acclimatization phase, the wild boars received metaphylactic treatment with oxytetracycline dihydrate (Alamycin LA 300, Norbrook Laboratories, Northern Ireland) and ivermectin (Ivomec S, Merial GmbH) in order to eliminate parasites and control inapparent bacterial infections. These wild boar piglets were not vaccinated and tested negative to the following main porcine pathogens in the region: Aujeszky virus, *Mycobacterium bovis, Mycoplasma pneumoniae,* and porcine circovirus type 2. Animal care and procedures were performed in accordance with the guidelines of the good experimental practice, following European, national, and regional regulations and under the supervision and approval of the Ethics Committee of the Comunidad de Madrid (reference PROEX 124/18 and 004/18).

### 2.3. Study Design

As stated above, the blood, oral fluid, and feces samples analyzed in the present study were obtained from two previous independent experiments.

In the first experiment, nine wild boars were orally inoculated with 10^4^ TCID_50_ of the attenuated Lv17/WB/Riel ASFV isolate, and three naïve wild boars were in direct contact with these orally inoculated animals (*n* = 12). Because of the high correlation between the presence of the viral DNA in shedding routes and the biological similarity of the infection route (oral and direct contact) in the orally inoculated and in-contact animals, we considered the nine orally inoculated and the three in-contact wild boars in a single group, denominated as the attenuated group. In the period of this experiment that is of interest for the current study, the vaccination period lasted 30 days [5].

In the second experiment, six wild boars were intramuscularly inoculated with 10 HAD_50_ of Arm07 ASFV isolate, and the remaining 11 wild boars were in direct contact with these intramuscularly inoculated animals (*n* = 17). No animals survived the infection. Because of the lack of differences with regard to clinical course, pathological findings, and virus presence in tissues (Appendix A) between the intramuscularly inoculated or in-contact animals, we considered the six wild boars intramuscularly inoculated and the 11 in-contact wild boars in a single group, denominated as the virulent group. Their study period lasted 15 days, as described previously by Rodríguez-Bertos et al. [22] (Figure 1).

### 2.4. Sampling and ASFV DNA Detection

All the animals from the two experiments were individually sampled twice a week and on the last day of their study period. Each wild boar was removed from the pen during sample collection and between successive samplings, and all the materials required for sampling were disinfected in order to avoid cross-contamination. Blood with ethylenediamine tetraacetic acid (EDTA) was collected from ophthalmic and cavernous sinuses at the medial angle of the eye behind the nictitating membrane [25], and these samples were processed immediately after their collection. Oral fluid and feces were collected using cotton swabs (Deltalab, Barcelona, Spain) and were stored at −80 °C until analysis. Negative control samples were collected on day 0.

DNA extraction was performed using the High Pure Template Preparation Mix Kit (Roche Diagnostics GmbH, Mannheim, Germany) according to the instructions for use. The detection of ASFV DNA in the different shedding routes was performed using the protocol of the TaqMan real-time quantitative PCR (qPCR) previously described by King et al. [26]. The oral fluid and feces samples were also analyzed to test for the presence of ASFV DNA using the UPL qPCR technique [27]. A positive result for the qPCR was determined by identifying the threshold cycle value (Ct) at which a reporter dye emission appeared above the background within 40 cycles. 

### 2.5. Statistical Analysis

The statistical analysis was conducted using SPSS 25 (IBM, Somar, NY, USA). The Ct values of the qPCR and the number of animals with positive quantitative PCR (qPCR) in different shedding routes were compared between groups and tests using a one-way ANOVA procedure. The Mann–Whitney U test was used to compare different virus administration routes and the starting time of positive results of the qPCR in blood, oral fluid, and feces samples in the two groups. Significance was considered at a *p*-value <0.05.

## 3. Results

### 3.1. Clinical Signs

All the animals from the virulent group developed clinical signs compatible with ASFV infection at 4 ± 1 days post infection (dpi) and succumbed to the disease at 10 ± 3 dpi. The main clinical signs observed in this group were high fever, lethargy, and anorexia. Some animals had slight erythema, ocular and nasal discharges, slight walking difficulties, and an increased respiratory rate. These observations were previously detailed by Rodríguez-Bertos et al. [22].

However, the animals from the attenuated group did not have any ASF-compatible clinical signs [5]. The only clinical reaction detected in this group was a slight increase in body temperature from 40.1 to 40.8 °C in eight of 12 animals (66.66%), which lasted 3.5 days (4–24 days post inoculation).

### 3.2. Inoculation and Contact Infection

A total of 592 individual paired samples of blood, oral fluid, and feces were analyzed, 361 from the attenuated group and 231 from virulent group. All the inoculated and in-contact animals were successfully infected in both independent experiments. The route of virus administration did not influence the Ct values obtained for the qPCR of the inoculated (orally or intramuscularly) and in-contact animals in either group (Mann–Whitney U test, *p* < 0.05). 

In the virulent group, all the animals attained positive qPCR results with regard to blood, while only eight out of 12 animals were qPCR-positive in the attenuated group (see Table 1). There was a significant difference between groups with regard to the number of animals with a positive qPCR in the case of blood (see Table 1; one-way ANOVA, F = 37.20, *p* < 0.05). We observed a similar number of positive animals in the case of ASFV DNA in oral fluid. However, the number of positive animals with regard to feces was higher in the virulent group than in the attenuated group (Mann–Whitney U test, Z = −3.85, *p* < 0.05). Additional information about antibody response in both groups is provided in Appendix A.

### 3.3. Detection of ASFV Genome and Starting Time of Shedding in Different Routes

The highest level of shedding was detected in the virulent group in blood (intramuscularly infected Ct = 27.91 ± 8.70; in-contact Ct = 31.05 ± 9.30), reaching the peak of viremia at 12 dpi (in-contact Ct = 16.32 ± 0.68). In the attenuated group, blood samples attained intermittently positive results with high Ct values for qPCR (orally infected Ct = 39.32 ± 2.64; in-contact Ct = 38.87 ± 2.51) (Figure 2). There was a significant difference between groups the Ct values of the qPCR in blood (see Table 2; one-way ANOVA, F = 36.13, *p* < 0.05). Only 11% of the blood samples analyzed for the attenuated group attained positive qPCR results (Figure 3). In the case of feces, ASFV DNA was occasionally detected. No significant differences between groups were detected in oral fluid samples (Figure 3). However, there was a significant difference between groups with regard to the Ct values of the qPCR in oral fluid and feces (Table 2; Figure 2; one-way ANOVA, F = 19.92; F = 36.06, *p* < 0.05).

Only a slight delay in the starting time of shedding was observed between the directly inoculated and contact animals in both groups. The animals from the virulent group started the shedding period earlier than animals from the attenuated group. On the first day of ASFV DNA detection, at least one animal from virulent group attained results of 3 dpi in blood, oral fluid, and feces. Otherwise, the attenuated group had more differences with regard to not only the starting time of shedding, but also the shedding routes. The attenuated isolate was detected for the first time at 10 dpi in oral fluid, 14 dpi in blood, and 21 dpi in feces. There was a significant difference between the two groups for the first time of detection of the ASFV genome in blood, oral fluid, and feces (Mann–Whitney *U* test; *Z* = −4.35, *Z* = −3.91, and *Z* = −3.77, respectively, *p* < 0.05). In addition, the detection of the ASFV genome in the virulent group was constant throughout the sampling period (from the first appearance until the day of death or euthanasia).

## 4. Discussion

The objective of the present study was to further evaluate the safety of the Lv17/WB/Rie1 ASFV isolate as a potential vaccine candidate in wild boar by evaluating the shedding patterns in comparison with the highly virulent Arm07 isolate. We observed significant differences between the shedding patterns of the animals from the attenuated and virulent groups. In general, fewer animals from the attenuated group were positive for the ASFV genome in blood, oral fluid, and feces when compared with those from the virulent group. Furthermore, the attenuated group started time of shedding later and with higher Ct values (above 39) when compared with the virulent group. All of this suggests that the attenuated ASFV isolate was excreted in smaller amounts or was a residual viral DNA.

Interestingly, in the attenuated group, the detection of viral DNA in oral fluid was slightly higher than in the other routes of excretion (see Figure 2). Moreover, the period of shedding in oral fluid was longer than in the case of blood and feces. This suggests that animals orally infected with an attenuated isolate are capable of containing a local infection at the entrance of the tonsils and lymph nodes, and they do not spread through viremia [28,29]. As one study pointed out, ASFV survival in oral fluid may be reduced by the potential presence of inhibitors in saliva [30]. This could possibly be why we were able to detect a viral genome in oral fluid, but the animals had neither clinical signs nor viremia. Another reason for this could be that the animals were kept in a contaminated environment during both experimental trials and could, therefore, have undergone several reinfections, but were protected and could control local oral replication. All of these reasons confirm that, despite the fact that oral fluid was a major shedding route in the attenuated group, it may have been overestimated in our experimental conditions. These observations, along with other results obtained in previous studies, support the importance of studying this route of excretion in ASFV infection, which may be useful as a noninvasive sampling strategy to detect an infected population and estimate environmental contamination [15,30,31].

Fecal samples were occasionally positive in the virulent group (29% of samples) and to an even lesser extent in the attenuated group (only 7% of samples). This can be explained by the fact that we did not observe hemorrhagic enteritis in those animals inoculated with either the virulent isolate or the attenuated isolate, which is, according to McVicar [32], the most likely cause of virus elimination through feces. Our results are similar to those of another study in which only 8.7% of the feces collected from viremic animals were positive [33]. In addition, it was previously observed that ASFV isolation in rectal samples rarely produced positive results [30]. ASFV shedding through feces seems to be limited, being more important in the acute phase of ASFV infections with highly virulent strains. Feces proved to be an infrequent route of excretions in the attenuated group, despite the oral inoculation route.

Furthermore, blood is traditionally known to be the main route of transmission of ASFV, because it contains a high virus load in infected animals [21,34]. This factor is especially important in wild boar populations because of fights among mating males, hunting activities, and scavenging behavior [35]. Interestingly, our study attained a low number of positive blood samples in the attenuated group throughout the study period (11%). Additionally, the viremia periods in this group were intermittent and had high qPCR Ct values (orally infected 39.32 ± 2.64). The virulent group had a higher number of positive blood samples (38%) with higher viremia (intramuscularly infected 27.91 ± 8.70), which is associated with a higher probability of collecting positive excretions [30]. All of this suggests that the risk of transmission through contact by contaminated blood is much lower with regard to infection with attenuated isolates when compared with natural infection with highly virulent isolates, and this is a valuable feature for safe vaccine candidates.

Virus isolation could be a complementary technique in this study to unravel the pattern shedding of ASFV. However, it was not possible to achieve satisfactory results with the virus isolation techniques available using established procedures [36] with this type of samples from wild boar. The methodology used to determine the viability of the ASFV included the viral isolation technique and the hemadsorption test in PBM. In addition, these tests have been assessed with PBMs from wild boar to increase sensitivity. However, both methods showed difficulties in development using these excretion samples from wild boar. Several authors argued that the difficulties in isolating ASFV from wild boar samples could be due to the poor quality of the samples of this species [37,38]. Our results reflect that the use of high-quality wild boar samples has not improved the success of ASFV isolation. This decreased sensitivity, both in viral isolation and in hemadsorption tests, with wild boar samples should be addressed in future works.

Regarding the incubation period, our results showed a slight difference in comparison to other studies carried out with domestic pigs [30,39]. One study where pigs were intramuscularly inoculated with Georgia 2007/1 showed that the incubation time lasted 5 days in blood, oral fluid, and feces [30]. Similar results were obtained in another study using three different isolates previously described by [39]. In the present study, we observed the first ASFV DNA detection in blood, oral fluid, and feces at 3 dpi in any animal from the virulent group, but the average number of the intramuscularly inoculated animals was 5 dpi in oral fluid and 8 dpi in blood and feces. The reasons for this could be that we used a lower infective dose and a different virulence of the ASFV isolate.

Despite the clearly significant differences observed between the animals infected with attenuated or virulent isolates, it is important to keep in mind that both groups were infected by different routes of administration, which could have influenced the results. However, the current study has not shown significant differences between intramuscularly or in-contact infected animals with regard to the shedding patterns related to the Ct values from the qPCR. Furthermore, no significant differences were observed in the shedding pattern of orally and in-contact infected animals from the attenuated group, beyond the slight delay of in-contact animals. This makes biological sense, because the animals in contact mimic a natural infection, as does the oral route of administration. These observations are according to other studies where naturally infected pigs showed a similar pattern in different routes of shedding to inoculated animals [15,30]. However, it is clear that the virulence and dose of the ASFV isolate influence the starting time and dynamic of shedding. Animals inoculated with highly virulent isolates present characteristic sub-acute clinical form of ASFV infection. Despite the important viremia, shedding through other routes can be rare in these animals.

It is not only important to take into account the level of viral excretions obtained from animals infected with attenuated or virulent isolates, but also to evaluate the environmental contamination of these excretions. Environmental contamination can play an important role in continuous infectious cycles, during which animals are constantly exposed to small amounts of ASFV originating from excretions. Previous studies confirmed that feed and bedding material (straw) contaminated with excretions from ASFV-infected pigs can be infectious for naïve pigs. In fact, the virus transmission from a contaminated environment to naïve pigs can occur only during a relatively short time period of only one day [40]. However, another study confirmed that infectious ASFV originating from animals infected with Georgia 2007/1 isolate could be detected for up to 3 days in feces and 5 days in urine at 21 °C [33]. More studies are, therefore, required evaluate environmental contamination and the kinetics of the different virus isolates in the wild boar population.

LAVs are currently the most promising and best positioned candidates for ASF control, including gene deletion mutants [6]. The study of the shedding patterns of different ASFV isolates is of great interest owing to the capacity of attenuated isolates to induce a high antibody response and the protection of vaccinated animals against infection with virulent ASFV Genotype II isolates [5]. In addition, a low shedding capacity of LAV candidates, which has also been described for the attenuated ASFV NH/P68 vaccine candidate [24], may help amplify vaccination coverage, thus reducing the need for the expensive production and large-scale administration of vaccine for wild boar in field conditions. This study demonstrates not only that the attenuated Lv17/WB/Rie1 ASFV isolate may be efficacious and safe as a vaccine candidate for the inoculated and in-contact wild boar, but also that the viral excretion from infected animals is limited and is relatively safe for the environment [41,42]. The current situation in the European Union (EU) and some eastern European countries indicates that the wild boar is the most severely affected host, thus conferring upon it an important role in ASF spread and maintenance [35]. Our study provides interesting information about shedding patterns of attenuated ASFV isolates such as Lv17/WB/Rie1, which is a promising vaccine candidate against ASF in wild boar.

## 5. Conclusions

In conclusion, this study provides valuable information on ASFV shedding patterns in wild boar. In general, fewer animals infected with the attenuated Lv17/WB/Rie1 isolate tested positive for ASFV in blood, oral fluid, and feces in comparison to animals infected with virulent Armenia07 isolate. Oral fluid has been shown to be the major shedding route in animals infected with attenuated ASFV isolate, and intermittent viremia periods with high qPCR Ct values were observed. ASFV shedding through feces rarely produced positive results in either group. This knowledge will help evaluate the shedding of new LAV candidates in wild boar populations, including the comparison with gene deletion mutant LAVs, whose current results are promising.

## Figures and Tables

**Figure 1 vaccines-08-00767-f001:**
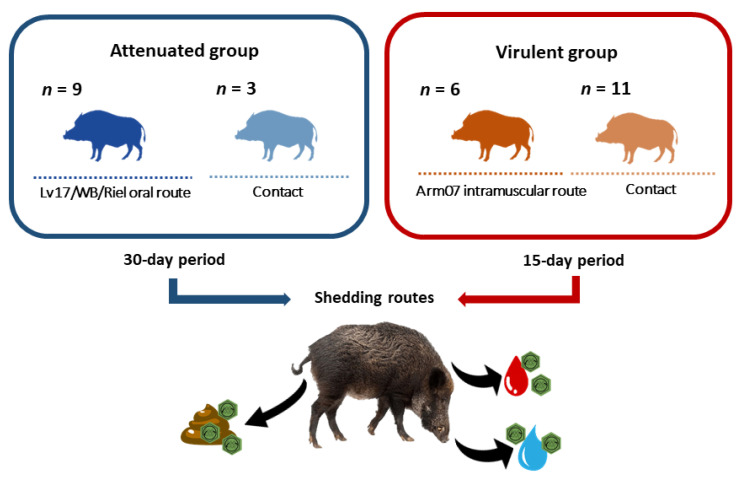
Schematic illustration of trial schedule and experimental groups. The attenuated group consisted of 12 animals (nine orally inoculated and three contacts). The virulent group consisted of 17 animals (six intramuscularly inoculated with Arm07 strain and 11 contacts). Three different shedding routes were evaluated (blood, oral fluid, and feces) in all the animals from both groups.

**Figure 2 vaccines-08-00767-f002:**
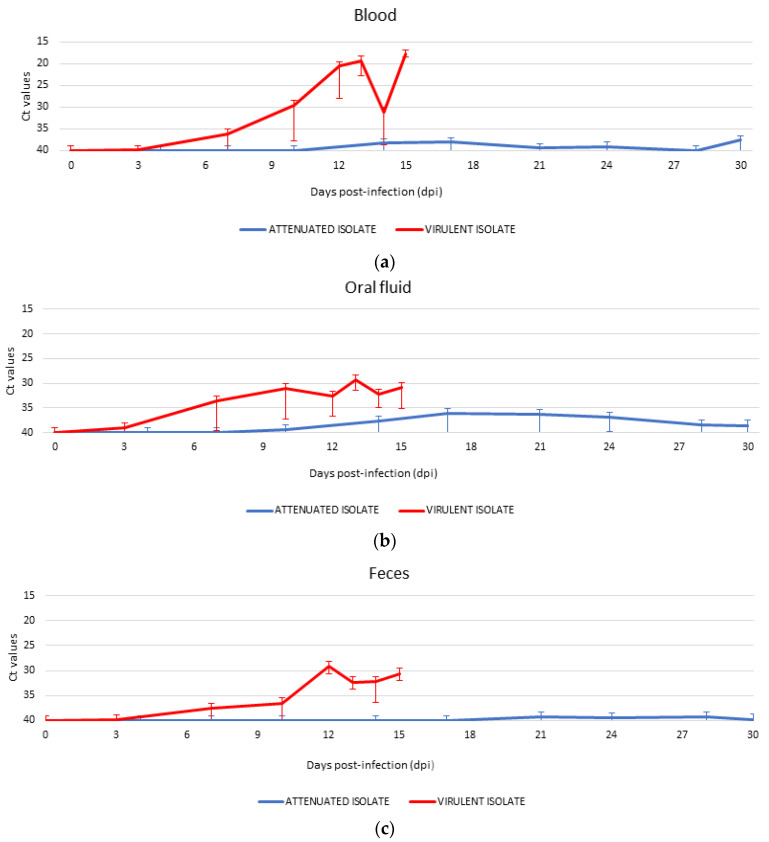
Results of viremia and virus shedding patterns observed in animals inoculated with Lv17/WB/Rie1 ASFV isolate (blue line) and inoculated with Arm07 ASFV isolate (red line): (**a**) ASFV DNA levels in blood; (**b**) ASFV DNA levels in oral fluid; (**c**) ASFV DNA levels in feces. Ct values from real-time PCR.

**Figure 3 vaccines-08-00767-f003:**
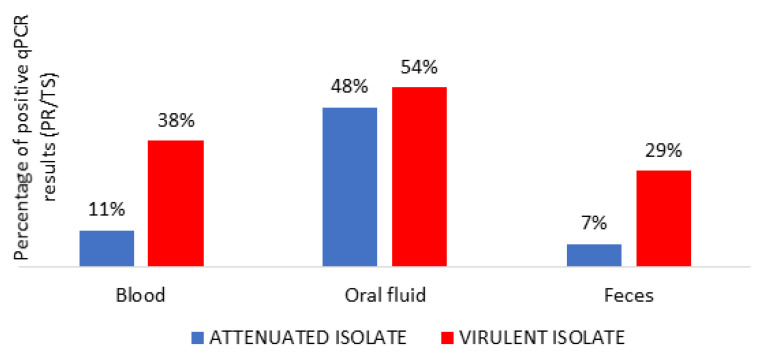
Percentage of positive real-time PCR (qPCR) results of ASFV DNA detection in blood, oral fluid, and feces in attenuated (blue) and virulent group (red). PR: positive results; TS: total number of samples.

**Table 1 vaccines-08-00767-t001:** Percentage of positive animals with real-time PCR results regarding African swine fever virus (ASFV) DNA detection in blood, oral fluid, and feces samples in attenuated and virulent groups during the 30 days of the experimental period.

Samples	Attenuated Group	Virulent Group
Blood	66%	100%
	8/12	17/17
Oral fluid	83%	88%
	10/12	15/17
Feces	25%	94%
	3/12	16/17

**Table 2 vaccines-08-00767-t002:** Results of the onset of ASFV DNA detection and Ct values of real-time PCR for Lv17/WB/Rie1 ASFV isolate in attenuated group and Arm07 in virulent group, during the 30 days of the experimental period.

Samples ^1^	Attenuated Group				Virulent Group			
	Orally Inoculated		In-Contact		Intramuscularly Inoculated		In-Contact	
	Ct Values of Real-Time PCR (±SD)	Days to Onset of ASFV DNA Detection ^2^	Ct Values of Real-Time PCR (±SD)	Days to Onset of ASFV DNA Detection ^2^	Ct Values of Real-Time PCR (±SD)	Days to Onset of ASFV DNA Detection ^2^	Ct Values of Real-Time PCR (±SD) in Contacts	Days to Onset of ASFV DNA Detection ^2^
**Blood**	39.32 (±2.64)	23 (±6)	38.87 (±2.51)	24	27.91 (±8.70)	8 (±3)	31.05 (±9.30)	10 (±2)
**Oral fluid**	38.49 (±3.67)	15 (±4)	37.31 (±3.60)	19 (±4)	30.08 (±4.77)	5 (±2)	32.42 (±5.55)	8 (±2)
**Feces**	39.87 (±0.76)	22 (±2)	39.39 (±1.81)	21	32.85 (±6.19)	8 (±4)	36.07 (±4.86)	12 (±2)

^1^ Results obtained by means of quantitative real-time polymerase chain reaction; ^2^ average number of days post infection of ASFV first DNA detection in the group (± standard deviation (SD)).

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
