# Peer review of "Distinct African Swine Fever Virus Shedding in Wild Boar Infected with Virulent and Attenuated Isolates"

_vaccines, 2020, doi:10.3390/vaccines8040767_

Round 1
Reviewer 1 Report
This ms describes an analysis of African swine fever virus (ASFV) shedding in wild boar piglets infected with either a wild-type strain(Arm/07) or a live attenuated virus strain (Lv17/WB/Rie1). Three sample types were compared: blood, oral secretions, and faeces. Findings seem unremarkable: shedding by the LAV is less than for the WT strain in blood and faeces although similar in oral secretions, and the time course of shedding is much faster with the WT strain. Methods appear to be adequate and data interpretations are reasonable. The writing is generally acceptable although here and there some prepositions are missing. A few particular issues are:
- It is not clear in the Abstract or Introduction, although made clear in the Study Design, that these animal studies were previously published by the same team and the present ms reports on samples collected in the course of those studies but not previously analyzed in detail. The present work is intended to complement those previous papers. For this purpose, the last paragraph of the Introduction should cite those papers (their references 5 and 23 if I am not mistaken).
- Line 180-181, blood samples for the LAV group, CT=33.07+/-5.04, peak CT 28.95+/-6.79. The first value is slightly different from that reported in Table 2 which shows CT=33.13+/-4.25. The second value (peak CT) is supported by Figure 3, but the viraemia data in the original publication (their reference #5, Figure 2) do not show any CT lower than about 34 prior to challenge. The inconsistencies are confusing and the authors must explain this more clearly.
- Table 2 is confusing in another respect: the columns, “onset of ASFV DNA detection,” report 2 sets of numbers, such as 21 (32 +/- 10) for faeces. According to the Legend, the first number is first day of detection following inoculation. Is this a mean value for all the piglets in the group, or first day of detection in any piglet of the group? The legend continues, “Days post infection on average +/-SD.” This must be the second value within the parentheses. Is this the mean total number of days of shedding? Or, the mean final day of shedding? In either case, since the entire sampling period is 30 days, it is not clear how the authors could have arrived at 32 +/- 10. The authors need to clarify in the Legend what those sets of numbers actually mean.
- In their analyses the authors combine together the boars that were inoculated with either the LAV or the WT, and the boars that were exposed by direct contact with those inoculated. Did they look for any differences in shedding pattern based on route of exposure?
- The references cited for the PCR assays (26, 27) suggest the methodology is adequate for specificity and sensitivity. However here the authors are comparing Ct values of samples obtained from different experiments. If the PCR testing was done at different times (as suggested by the different sample origins), what controls such as standard positive control samples were included to normalize the results and assure that the PCR assays were behaving similarly for both sample sets? Or, were all samples tested side-by-side?
Author Response
This ms describes an analysis of African swine fever virus (ASFV) shedding in wild boar piglets infected with either a wild-type strain(Arm/07) or a live attenuated virus strain (Lv17/WB/Rie1). Three sample types were compared: blood, oral secretions, and faeces. Findings seem unremarkable: shedding by the LAV is less than for the WT strain in blood and faeces although similar in oral secretions, and the time course of shedding is much faster with the WT strain. Methods appear to be adequate and data interpretations are reasonable. The writing is generally acceptable although here and there some prepositions are missing. A few particular issues are:
We would like to thank the reviewer for taking the time for reviewing this manuscript and for providing a clear, constructive, and detailed changes and suggestions, which were carefully considered and modified in the new version of the manuscript.
- It is not clear in the Abstract or Introduction, although made clear in the Study Design, that these animal studies were previously published by the same team and the present ms reports on samples collected in the course of those studies but not previously analyzed in detail. The present work is intended to complement those previous papers. For this purpose, the last paragraph of the Introduction should cite those papers (their references 5 and 23 if I am not mistaken).
We agree. As suggested, the present work aims to complement the shedding characterization of previous studies carried out by our team regarding ASFV infection in wild boar. Accordingly, we have clarified and added this information and cited those studies in the last paragraph of the introduction. Thank you for this suggestion.
This paper examines the shedding patterns of the attenuated ASFV isolate Lv17/WB/Riel in order to evaluate its potential as a vaccine candidate for wild boar. The possible survival and shedding of AFSV was investigated by collecting blood, oral swabs and feces while carrying out previously published studies in which wild boar were orally inoculated with the Lv17/WB/Rie1 ASFV isolate [5] and other study where wild boar were intramuscularly inoculated with the highly virulent Arm07 ASFV isolate [23]. This information will help to determine the potential risk of the deliberate release into the environment of this attenuated isolate in comparison with highly virulent field isolates, and will additionally complement studies previously described by Barasona et al. [5] and Rodríguez-Bertos et al. [23].
- Line 180-181, blood samples for the LAV group, CT=33.07+/-5.04, peak CT 28.95+/-6.79. The first value is slightly different from that reported in Table 2 which shows CT=33.13+/-4.25. The second value (peak CT) is supported by Figure 3, but the viraemia data in the original publication (their reference #5, Figure 2) do not show any CT lower than about 34 prior to challenge. The inconsistencies are confusing and the authors must explain this more clearly.
We thank the reviewer for this appreciation and careful comparison between the results of the graphs and tables. Thanks to that we have been able to detect an accumulated bias in one part of our converged database that to solve it we have had to modify all the Ct results, and all the graphs of figure 3 as well. New statistical analysis indicates that there were differences in Ct between blood, oral fluid and feces. Marked changes throughout the manuscript.
- Table 2 is confusing in another respect: the columns, “onset of ASFV DNA detection,” report 2 sets of numbers, such as 21 (32 +/- 10) for faeces. According to the Legend, the first number is first day of detection following inoculation. Is this a mean value for all the piglets in the group, or first day of detection in any piglet of the group? The legend continues, “Days post infection on average +/-SD.” This must be the second value within the parentheses. Is this the mean total number of days of shedding? Or, the mean final day of shedding? In either case, since the entire sampling period is 30 days, it is not clear how the authors could have arrived at 32 +/- 10. The authors need to clarify in the Legend what those sets of numbers actually mean.
We agree. It is confusing. We have decided to modify values from column ¨days to onset of ASFV DNA detection¨. First value is the average of days post infection of first ASFV DNA detection in the group (+/- standard deviation). Values of the onset of ASFV detection in blood/oral fluid/feces in any piglet of the group were described in results (Section 3.3.).
We apologize for this confusion. Initially, we have analyzed data from 54 days of experimental period (study described by Barasona et al., 2019), but only the first 30 days of vaccinated period corresponding to the shedding patterns of the attenuated isolate.
The reason why we have decided to evaluate the shedding patterns of attenuated ASFV isolate only during the first 30 days of the experiment (before challenge) was to avoid a potential interference in shedding results influenced by the virulent isolate of the challenge.
Due to this confusion, we have revised the entire manuscript in this regard. Results of ASFV DNA from qPCR are correct, however the onset of ASFV DNA detection and number of positive animals in attenuated group (Table 1) have changed. We removed data of animals which started with shedding after 30 days, because it was a result of virus challenge introduction. For this reason, values of positive animals from attenuated group (in oral fluid and feces; Table 1) have changed.
- In their analyses the authors combine together the boars that were inoculated with either the LAV or the WT, and the boars that were exposed by direct contact with those inoculated. Did they look for any differences in shedding pattern based on route of exposure?
In-contact animals started the virus shedding sightly later compared with the orally or intramuscularly inoculated. This delay is due to the incubation period since a natural infection. This difference was also described in virulent group for onset of clinical signs and the moment of death, described by Rodríguez-Bertos et al. 2020.
However, statistical analysis confirmed that the route of virus administration does not influence in Ct values from qPCR between the inoculated (orally or intramuscularly) and in-contact animals from both groups.
We totally agree that this information has to be taken into account by the reader when evaluating the results. Therefore, we want to make it clearer and we added additional information to Section 3.2. Inoculation and contact infection and in the discussion:
Line 175-177: The route of virus administration did not influence in the Ct values obtained for the qPCR of the inoculated (orally or intramuscularly) and in-contact animals in either group (Mann-Whitney U test, p < 0.05).
Despite the clearly significant differences observed between the animals infected with attenuated or virulent isolates, it is important to keep in mind that both groups were infected by different routes of administration, which could have influenced the results. However, the current study has not shown significant differences in between intramuscularly or in-contact infected animals as regards the shedding patterns related to the Ct values from the qPCR. Furthermore, no significant differences were observed in the shedding pattern of orally and in-contact infected animals from the attenuated group, beyond the slight delay of in-contact animals. This makes biological sense, because the animals in contact mimic a natural infection, as does the oral route of administration.
- The references cited for the PCR assays (26, 27) suggest the methodology is adequate for specificity and sensitivity. However here the authors are comparing Ct values of samples obtained from different experiments. If the PCR testing was done at different times (as suggested by the different sample origins), what controls such as standard positive control samples were included to normalize the results and assure that the PCR assays were behaving similarly for both sample sets? Or, were all samples tested side-by-side?
Right. As mentioned before, collected samples were taken from two complementary experiments. This is due to the logistical limitation of working in BSL3 conditions, the difficulties of handing these ¨wild¨ animals, and the maximum reduction of the number of animals used per experiment to meet animal welfare requirements. However, analysis of all samples for this study were performed at same time to compare result for each shedding route. In addition, all analysis were carried out in the Reference Laboratory for the World Organization for Animal Health (OIE) for ASF. We always use standardized protocols and well-known sample (with low Ct value and previously sequenced) as ASF positive control in qPCR assay. This allows us to validate the qPCR assay and compare Ct values coming from different studies.

Reviewer 2 Report
Comments on Distinct African swine fever virus shedding in wild boar infected with virulent and attenuated isolates
In this study, Aleksandra Kosowska et al. examined the shedding patterns in wild boar infected with either a virulent or an attenuated ASFV strain. The work is helpful for epidemiological surveillance and disease control. But several major concerns should be addressed and clarified.
Major issues:
- Figure 1 is confusing, especially for the attenuated group. “After 30 days expose to:”; “4 naïve wild boars challenged with high virulent Arm07”… Are these statements correct?
- The incubation time and onset of virus shedding in the infected pigs should be provided and compared.
- The virus shedding through various routes and virus distributions in different tissues should also be evaluated by virus isolation, which is more clinically relevant than real-time PCR.
- The antibody titers induced by the challenge virus should be tested.
- The manuscript should be improved by native English speakers.
Minor points:
- The inoculation routes of the two experiments are different. This should be justified.
- In the two experiments, there is no significant difference in shedding patterns between the inoculated and the in-contact wild boar. This should be explained.
- Section 2.5, the number of repeated tests of the samples needs to be provided.
- Line 32, replace “The virus” with “ASFV”.
- Line 40, replace “vaccine” with “ASF vaccine”.
- Line 89 and elsewhere, replace “TCID50/ml” with “TCID50/ml”.
Author Response
In this study, Aleksandra Kosowska et al. examined the shedding patterns in wild boar infected with either a virulent or an attenuated ASFV strain. The work is helpful for epidemiological surveillance and disease control. But several major concerns should be addressed and clarified.
We would like to thank the reviewer for positive appreciation and for taking the time for reviewing this manuscript and for providing a constructive and detailed changes to improve in clarity, which were carefully considered and modified in the new version of the manuscript.
Major issues:
- Figure 1 is confusing, especially for the attenuated group. “After 30 days expose to:”; “4 naïve wild boars challenged with high virulent Arm07”… Are these statements correct?
Thank you for raising such an important point. We totally agree. There has been a mistake when making the figure since data for attenuated group come from a more complex study. This has been clarified at the end of the introduction section:
This information will help to determine the potential risk of the deliberate release into the environment of this attenuated isolate in comparison with highly virulent field isolates, and will additionally complement studies previously described by Barasona et al. [5] and Rodríguez-Bertos et al. [23].
The first study described by Barasona et al., 2019, was focused on the clinical trial vaccination and challenge. However, for the current study of the shedding patterns, we are only interested on vaccination period to avoid a potential interference in shedding results influenced by the virulent isolate of the challenge. This period includes the first 30 days of the experiment described in Barasona et al. 2019, before challenge. After that, vaccinated animals were exposed to four wild boar intramuscularly infected with virulent ASFV isolate, Armenia07. According to your suggestion, we have modified the figure.
In order to clarify this point, we have added a sentence in “study design” section: “The period of this experiment that interests us for the current study, the vaccination period, lasted 30 days.”
- The incubation time and onset of virus shedding in the infected pigs should be provided and compared.
We agree. Thank you for this suggestion. Incubation time and onset of virus shedding have been included in the discussion section to support our results. We have added the following citation to the text:
Regarding the incubation period, our results showed a slight difference in comparison to other studies carried out with domestic pigs [30, 41]. One study where pigs were intramuscularly inoculated with Georgia 2007/1 showed that the incubation time lasted 5 days in blood, oral fluid and feces [30]. Similar results were obtained in another study using three different isolates previously described by [41]. In present study, we observed first ASFV DNA detection in blood, oral fluid and feces at 3 dpi in any animal from the virulent group. However, the average number of the group was 9 dpi in blood, 8 dpi in oral fluid and 11 dpi in feces. The reasons for this could be that we have used a lower infective dose and a different virulence of the ASFV isolate.
- The virus shedding through various routes and virus distributions in different tissues should also be evaluated by virus isolation, which is more clinically relevant than real-time PCR.
Thank you for your suggestion. We agree that virus isolation is an important technique to evaluate the presence of the infectious virus in different excretions. In fact, it was one important objective for this study. However, it was not possible due to limitations due to pandemic situation, and limited access to the BSL3 facilities during the last year. So, we were able to analyze only a reduced number of selected samples.
ASFV isolation of attenuated and virulent isolate was not possible after three passages in PBM cells, neither characteristic hemadsorption ASFV pattern for virulent isolate was observed.
Previous experience shows that ASFV isolation from wild boar samples is complicated, even despite the high values of viral DNA detected according to Ct values. We believe that qPCR assay is extremely useful for ASFV diagnosis, because allows to detect a highly conserved structural protein, p72 and it would perform consistently with both hemadsorbing and nonhemadsorbing isolates.
A previous study described by Gallardo et al., 2015 confirmed a relatively small number of successfully isolated wild boars’ samples, where only 30.7% were isolated compared with 86% of samples from domestic pigs.
Gallardo C, Nieto R, Soler A, Pelayo V, Fernández-Pinero J, Markowska- Daniel I, Pridotkas G, Nurmoja I, Granta R, Simón A, Pérez C, Martín E, Fernández- Pacheco P, Arias M. 2015. Assessment of African swine fever diagnostic techniques as a response to the epidemic outbreaks in eastern European Union countries: how to improve surveillance and control programs. J Clin Microbiol 53:2555–2565.
Virus distribution in different tissues was evaluated by qPCR in two previous studies described by Barasona et al., 2019 and Rodríguez-Bertos et al., 2020.
- The antibody titers induced by the challenge virus should be tested.
We tested serum samples for the presence of ASFV-specific antibodies using commercial ELISA kit for the detection of p72 ASFV antigen, Ingenasa-Ingezim PPA Compac K3.
The virulent group consist of 17 wild boar; six of them was intramuscularly infected with the virulent ASFV isolate and antibody response was not detected in these animals; in four of eleven virulent isolate in-contact animals the presence of ASFV-specific antibodies was detected in the last day of sampling, just before death/euthanasia. In two of them also in the previous sampling at 13 days post-infection.
Summarizing, antibody response in animals from the virulent group was not protective and appeared too late to significantly reduce the amount of excreted virus through different shedding routes. For this reason, we have considered these results not relevant for the evaluation of virulent ASFV isolate shedding patterns and not including them in the manuscript.
- The manuscript should be improved by native English speakers According to suggestions, a final version of the entire manuscript have been edited to improve English language by a professional native speaker.
Minor points:
- The inoculation routes of the two experiments are different. This should be justified.
We have used information proportionated for previous studies, as explained in the manuscript. These previous studies had their scientific reasons for choosing these routes of administration. We agree that intramuscular route could influence the results comparing with the oral route of administration. However, despite this and based on the bibliography, we considered that this potential influence on our comparative results does not represent a significant bias.
Howey EB, O’Donnell V, de Carvalho Ferreira HC, Borca MV, Arzt J: Pathogenesis of highly virulent African swine fever virus in domestic pigs exposed via intraoropharyngeal, intranasopharyngeal, and intramuscular inoculation, and by direct contact with infected pigs. Virus Res 2013, 178:328–339.
Furthermore, statistical analysis confirmed that the route of virus administration does not influence in Ct values from qPCR between the inoculated (orally or intramuscularly) and in-contact animals from both groups. This information was added to Section 3.2. Inoculation and contact infection.
But we totally agree that this information has to be taken into account by the reader when evaluating the results. Therefore, we want to make it clearer in the discussion:
Despite the clearly significant differences observed between the animals infected with attenuated or virulent isolates, it is important to keep in mind that both groups were infected by different routes of administration, which could have influenced the results. However, the current study has not shown significant differences in between intramuscularly or in-contact infected animals as regards the shedding patterns related to the Ct values from the qPCR. Furthermore, no significant differences were observed in the shedding pattern of orally and in-contact infected animals from the attenuated group, beyond the slight delay of in-contact animals. This makes biological sense, because the animals in contact mimic a natural infection, as does the oral route of administration.
- In the two experiments, there is no significant difference in shedding patterns between the inoculated and the in-contact wild boar. This should be explained.
We totally agree. For this reason, the manuscript has been modified with extra information (as was previously explained in Minor points, 1).
- Section 2.5, the number of repeated tests of the samples needs to be provided. It has been added and clarified.
- Line 32, replace “The virus” with “ASFV”. It changed.
- Line 40, replace “vaccine” with “ASF vaccine”. It changed.
- Line 89 and elsewhere, replace “TCID50/ml” with “TCID50/ml”. It changed.

Round 2
Reviewer 2 Report
Comments for Distinct African swine fever virus shedding in wild boar infected with virulent and attenuated isolates (1011525-peer-review-v2)
The authors have made comprehensive revisions according to the comments, The manuscript has been improved significantly. However, several important issues remain to be addressed.
- Considering the different inoculation routes and doses of the virulent and attenuated ASFV strains, it seems unjustified to conduct such a comparison.
- In addition to real-time PCR, virus isolation seems to be more useful.
- Anti-ASFV antibodies and ASFV genomic DNA in various organs of all the wild boar should be shown, especially for the in-contact wild boar in the attenuated group.
- The results for the orally/intramuscularly inoculated and contacted wild boar should be presented respectively.
- In the two experiments, there is no significant difference in shedding patterns between the inoculated and the in-contact wild boar. This should be discussed.
Author Response
The authors have made comprehensive revisions according to the comments, The manuscript has been improved significantly. However, several important issues remain to be addressed.
- Considering the different inoculation routes and doses of the virulent and attenuated ASFV strains, it seems unjustified to conduct such a comparison.
As we detailed in the manuscript, we have used two main ways of infection due to the particular conditions of working with this virus in wild boar (extrapolated to field conditions) and to the previous experiences reflected and agreed in the scientific literature with animal trials (IM challenge). Therefore, an intramuscular challenge in the control animals (only six), put in direct contact with 11 wild boars, simulating the oral/natural infection that we have also carried out with the oral inoculations in the attenuated group. This fact has been discussed in the manuscript according to point 5 of this revision letter. In addition, we have incorporated all the separate data for these subgroups, based on point 4 of this revision, In this sense, a new Table 2 has been provided in the manuscript, where no significant differences in results were found among groups.
- In addition to real-time PCR, virus isolation seems to be more useful.
We agree that virus isolation could be a complementary technique in this study. However, as explained in the first review, it was not possible to achieve satisfactory results with the available techniques of virus isolation in the laboratory of the UCM or in the CISA-INIA (European Reference Laboratory of ASF) with this type of samples from wild boar. Unfortunately, with the wild boar samples we are not able to isolate the ASFv, as we do with domestic pig samples, where we use PBM cells from domestic pig donors. The methodology used to determine the ASFv viability included viral isolation technique and the hemadsorption test, even with PBM from wild boar to increase sensibility. However, both methods presented difficulties in their development using these samples. Several authors argued that the difficulties for ASFv isolation from wild boar samples might be due to the poor quality of the samples from this species. Anyway, our results reflect that the use of high-quality samples from wild boar has not improved the success of the ASFv isolation. Thus, the methodology used here to achieve progress in isolations with peripheral macrophages from donor pigs and wild boar was not significantly effective.
In addition, the use of samples from excretions, oral and rectal samples, drastically reduces further conclusive results derived from viral isolation, due to possible contamination, or treatment by antibiotics.
This table shows the reduced number of isolations obtained in this study on a subsample. Obviously, this information lacks diagnostic value, mainly because the animals presented a high load of ASFV in most of the obtained samples, owing to the infection was clearly developed in the control group (virulent isolate).
|
GROUP |
DPI |
SAMPLE |
Original CT value |
1 |
2 |
3 |
HAD |
Final results |
|
Attenuated |
24 |
oral fluid |
37,51 |
41,88 |
Neg |
Neg |
no |
NEG |
|
Attenuated |
7 |
oral fluid |
32,94 |
Neg |
Neg |
Neg |
no |
NEG |
|
Attenuated |
21 |
oral fluid |
29,54 |
Neg |
Neg |
Neg |
no |
NEG |
|
Attenuated |
30 |
feces |
36,16 |
Neg |
Neg |
Neg |
no |
NEG |
|
Attenuated |
21 |
oral fluid |
30,57 |
Neg |
Neg |
Neg |
no |
NEG |
|
Attenuated |
11 |
oral fluid |
30,38 |
41 |
Neg |
Neg |
no |
NEG |
|
Attenuated |
25 |
oral fluid |
36,4 |
Neg |
Neg |
Neg |
no |
NEG |
|
Attenuated |
25 |
feces |
38,22 |
Neg |
Neg |
Neg |
no |
NEG |
|
Attenuated |
28 |
oral fluid |
33,07 |
43,01 |
Neg |
Neg |
no |
NEG |
|
Attenuated |
7 |
oral fluid |
38,03 |
Neg |
39,94 |
Neg |
no |
NEG |
|
Attenuated |
11 |
oral fluid |
29,01 |
Neg |
Neg |
Neg |
no |
NEG |
|
Attenuated |
11 |
feces |
32,34 |
40,85 |
36,13 |
Neg |
no |
NEG |
|
Attenuated |
25 |
oral fluid |
35,21 |
Neg |
Neg |
Neg |
no |
NEG |
|
Attenuated |
25 |
feces |
28,55 |
Neg |
37,81 |
Neg |
no |
NEG |
|
Attenuated |
24 |
oral fluid |
31,37 |
Neg |
Neg |
Neg |
no |
NEG |
|
Attenuated |
18 |
blood |
32,14 |
|
|
|
no |
POS no HAD |
|
Virulent |
4 |
oral fluid |
31,3 |
Neg |
Neg |
Neg |
no |
NEG |
|
Virulent |
4 |
feces |
35,93 |
Neg |
Neg |
Neg |
no |
NEG |
|
Virulent |
7 |
oral fluid |
29,74 |
36,62 |
Neg |
Neg |
no |
NEG |
|
Virulent |
7 |
feces |
19,39 |
32,21 |
Neg |
Neg |
no |
NEG |
|
Virulent |
4 |
oral fluid |
33,99 |
36,09 |
Neg |
Neg |
no |
NEG |
|
Virulent |
4 |
feces |
29,69 |
Neg |
38,31 |
Neg |
no |
NEG |
|
Virulent |
5 |
oral fluid |
24,05 |
Neg |
38,51 |
Neg |
no |
NEG |
|
Virulent |
5 |
feces |
24,1 |
37,24 |
41,46 |
Neg |
no |
NEG |
|
Virulent |
12 |
blood |
14,68 |
31,58 |
39,7 |
Neg |
no |
NEG |
|
Virulent |
15 |
blood |
24,2 |
38,59 |
37,83 |
Neg |
no |
NEG |
|
Virulent |
12 |
blood |
29,41 |
Neg |
Neg |
Neg |
no |
NEG |
|
Virulent |
12 |
blood |
18,62 |
Neg |
38,01 |
Neg |
no |
NEG |
|
Virulent |
15 |
blood |
30,78 |
Neg |
38,23 |
Neg |
no |
NEG |
|
Virulent |
8 |
blood |
22,2 |
37,94 |
Neg |
Neg |
no |
NEG |
|
Virulent |
6 |
blood |
27,18 |
|
|
|
yes |
POS HAD |
|
Virulent |
9 |
blood |
15,21 |
|
|
|
yes |
POS HAD |
We have now incorporated this statement in the discussion section:
Virus isolation could be a complementary technique in this study to unravel the pattern shedding of ASFV. However, it was not possible to achieve satisfactory results with the virus isolation techniques available using established procedures (OIE 2019) with this type of samples from wild boar. The methodology used to determine the viability of the ASFV included the viral isolation technique and the hemadsorption test in porcine blood
monocytes (PBM). In addition, these tests have been assessed with PBM from wild boar to increase sensitivity. However, both methods showed difficulties in development using these excretion samples from wild boar. Several authors argued that the difficulties in isolating ASFV from wild boar samples could be due to the poor quality of the samples of this species (Gallardo et al., 2015; Mazur-Panasiuk et al., 2019). Our results reflect that the use of high quality wild boar samples has not improved the success of ASFV isolation. This decreased sensitivity, both in viral isolation and hemadsorption tests, with wild boar samples should be addressed in future works.
OIE (2019) Manual of diagnostic tests and vaccines for terrestrial animals [online]. Paris, France: OIE. African swine fever. Retrieved from https://www.oie.int/en/standard-setting/terrestrial-manual/access-online/
Gallardo C, Nieto R, Soler A, Pelayo V, Fernández-Pinero J, Markowska- Daniel I, Pridotkas G, Nurmoja I, Granta R, Simón A, Pérez C, Martín E, Fernández- Pacheco P, Arias M. 2015. Assessment of African swine fever diagnostic techniques as a response to the epidemic outbreaks in eastern European Union countries: how to improve surveillance and control programs. J Clin Microbiol 53:2555–2565.
Mazur-Panasiuk, N., Woźniakowski, G., & Niemczuk, K. (2019). The first complete genomic sequences of African swine fever virus isolated in Poland. Scientific reports, 9(1), 1-9.
- Anti-ASFV antibodies and ASFV genomic DNA in various organs of all the wild boar should be shown, especially for the in-contact wild boar in the attenuated group.
OK, we understand that this information can be interesting for the readers. For this reason, we considered that summarized results of ASFV DNA detection from qPCR in tissues and additional information about antibodies should be uploaded as a Supplementary Material. Also, additional explanation was provided below.
Results of ASFV DNA from qPCR and virus isolation from tissues in attenuated group (orally infected and in-contact animals) were previously described by Barasona et al. 2019 (Outcomes after challenge Section; Figure 5; Figure 6).
In this case, sixteen different tissues have been analyzed. As we can read in Results Section, in three of eight orally infected and in two of three in-contact animals ASFV DNA was not detected in any of 16 tissues.
The remaining animals that survived showed weakly positive PCR results (Ct = 38.416 ± 1.16) in an average of 5 tissues. ASFV could be isolated from only two of 22 tissues analyzed from these animals: retropharyngeal lymph node in one vaccinated animal, and renal lymph node in one VContact animal. These virus isolates were non-hemadsorbing. The two late in-contact wild boar showed weakly positive PCR results in 9 or 12 tissues (Ct = 37.40 ± 0.43), where hemadsorbing ASFV was isolated only from one inguinal lymph node. (…) the one orally vaccinated animal that remained unprotected against challenge showed strongly positive PCR results (Ct = 23.32 ± 1.60) in all 16 tissues tested.
Results of ASFV DNA from qPCR in virulent group (intramuscularly inoculated and in-contact animals) were previously described by Rodríguez-Bertos et al. 2020 (Table 1). Blood as tissue sample was included in the Table 1, but these data come from the last blood sampling of dead animal.
In the study Rodríguez-Bertos et al. (Section 2.3. Tissue Sample Analysis) all details have been described.
In all of the animals from the two different groups (IM inoculated and in-contact animals),
ASFV DNA was detected in all 18 tissues evaluated. The cycle threshold (CT) values did not differ significantly between IM-inoculated or in-contact infected animals (Mann-Whitney U test, p = 0.331)
Results of blood presented in the manuscript include average Ct values of real-time PCR for the group during experimental period.
|
Tissue |
Attenuated group |
Virulent group |
||
|
Orally infected animals (n=9) |
In-contact animals (n=3) |
Intramuscularly infected (n=6) |
In-contact animals (n=11) |
|
|
Heart |
37 ± 5 |
37 ± 2 |
25 ± 7 |
23 ± 3 |
|
Lung |
37 ± 5 |
Neg |
22 ± 3 |
22 ± 2 |
|
Kidney |
38 ± 5 |
Neg |
24 ± 5 |
22 ± 2 |
|
Liver |
38 ± 4 |
Neg |
23 ± 7 |
20 ± 2 |
|
Urinary bladder |
37 ± 5 |
Neg |
26 ± 6 |
26 ± 3 |
|
Spleen |
36 ± 6 |
39 ± 2 |
22 ± 6 |
20 ± 4 |
|
Lymph nodes* |
37 ± 1 |
39 ± 1 |
25 ± 5 |
23 ± 2 |
|
Tonsil |
no data |
no data |
25 ± 8 |
22 ± 7 |
|
Bone Marrow |
38 ± 6 |
Neg |
24 ± 8 |
22 ± 4 |
|
Brain |
38 ± 5 |
Neg |
25 ± 7 |
23 ± 2 |
*Lymph nodes: These CT values were obtained from a pool of eight different lymph nodes: renal, mediastinal, retropharyngeal, mesenteric, preescapular, gastrohepatic, inguinal and mandibular lymph nodes.
Regarding antibody detection in blood, we tested serum samples for the presence of ASFV-specific antibodies using commercial ELISA kit for the detection of p72 ASFV antigen, Ingenasa-Ingenzim PPA Compac K3. Detailed information for the attenuated group was previously described by Barasona et al. 2019. In this figure, we show antibody response in both groups:
- The results for the orally/intramuscularly inoculated and contacted wild boar should be presented respectively.
Thank you for your suggestion. Despite no significant differences in Ct values results from qPCR between groups, we added extra columns to present separately results for orally infected and contacts animals in attenuated group and for intramuscularly infected and contacts animals in virulent group. Table 2 has been modified in the manuscript.
|
|
Attenuated group
|
Virulent group |
|||||||
|
Orally infected |
In-contact animals |
Intramuscularly infected |
In-contacts animals |
||||||
|
Ct values of real-time PCR (± SD) |
Days to onset of ASFV DNA detection |
Ct values of real-time PCR (± SD) |
Days to onset of ASFV DNA detection |
Ct values of real-time PCR (± SD) |
Days to onset of ASFV DNA detection |
Ct values of real-time PCR (± SD) in contacts |
Days to onset of ASFV DNA detection |
||
|
Blood |
39.32 (± 2.64) |
23 (± 6) |
38.87 (± 2.51) |
24 |
27.91 (± 8.70) |
8 (± 3) |
31.05 (± 9.30) |
10 (± 2) |
|
|
Oral fluid |
38.49 (± 3.67) |
15 (± 4) |
37.31 (± 3.60) |
19 (± 4) |
30.08 (± 4.77) |
5 (± 2) |
32.42 (± 5.55) |
8 (± 2) |
|
|
Feces |
39.87 (± 0.76) |
22 (± 2) |
39.39 (± 1.81) |
21 |
32.85 (± 6.19) |
8 (± 4) |
36.07 (± 4.86) |
12 (± 2) |
|
- In the two experiments, there is no significant difference in shedding patterns between the inoculated and the in-contact wild boar. This should be discussed.
We agree. Previously the manuscript has been improved with following paragraph to the Discussion:
Despite the clearly significant differences observed between the animals infected with attenuated or virulent isolates, it is important to keep in mind that both groups were infected by different routes of administration, which could have influenced the results. However, the current study has not shown significant differences in between intramuscularly or in-contact infected animals as regards the shedding patterns related to the Ct values from the qPCR. Furthermore, no significant differences were observed in the shedding pattern of orally and in-contact infected animals from the attenuated group, beyond the slight delay of in-contact animals. This makes biological sense, because the animals in contact mimic a natural infection, as does the oral route of administration.
Our results are in line with others studies where in-contact animals showed similar shedding patterns to inoculated animals, with difference in time to onset of ASFV DNA detection between these groups of animals:
Guinat, C.; Reis, A. L.; Netherton, C. L.; Goatley, L.; Pfeiffer, D. U.; Dixon, L. Dynamics of African Swine Fever Virus Shedding and Excretion in Domestic Pigs Infected by Intramuscular Inoculation and Contact Transmission. 2014, 1–9.
de Carvalho Ferreira, H. C.; Weesendorp, E.; Elbers, A. R. W.; Bouma, A.; Quak, S.; Stegeman, J. A.; Loeffen, W. L. A. African Swine Fever Virus Excretion Patterns in Persistently Infected Animals: A Quantitative Approach. Vet. Microbiol. 2012, 160 (3–4), 327–340. https://doi.org/10.1016/j.vetmic.2012.06.025.
In order to clarify this point, we have added the following paragraph to the Discussion:
These observations are according to other studies where naturally infected pigs showed a similar pattern in different routes of shedding to inoculated animals [15, 30]. However, it is clear that the virulence and dose of the ASFV isolate influence in starting time and dynamic of shedding. Animals inoculated with highly virulent isolates present characteristic sub-acute clinical form of ASFV infection. Despite the important viraemia, shedding through other routes can be rare in these animals.